# Secondary School Apprenticeship Research Experience: Scientific Dispositions and Mentor-Student Interaction

**Mercedes Edry [1], Irit Sasson [1,2,\*] and Yehudit Judy Dori [3,4]**

[1] Shamir Research Institute, University of Haifa, Haifa 3498838, Israel; merchied29@gmail.com
[2] The Department of Education, Tel-Hai College, Upper Galilee 1220800, Israel
[3] The Faculty of Education in Science and Technology, Technion-Israel Institute of Technology, Haifa 3200003, Israel; yjdori@ed.technion.ac.il
[4] The Samuel Neaman Institute for National Policy Research, Haifa 3200003, Israel
[\*] Correspondence: iritsa@telhai.ac.il

**Abstract:** This study investigated the impact of a secondary school science, technology, engineering, and mathematics (STEM) research apprenticeship program (STEM-RAP) as part of active learning pedagogy on students' performance. We examined students' (a) scientific dispositions—self-efficacy, intrinsic goal orientation, and sense of control over learning, (b) STEM career choice, and (c) mentor-student interaction. Research tools included open- and closed-ended questionnaires, as well as interviews with a sample of students and mentors. The questionnaire was administered to 319 11th and 12th grade students majoring in science and technology in Israeli high schools. Of these, 262 participated in STEM-RAP and 57 took part only in studying a high-school STEM subject as a major. The results show highly positive scientific dispositions. A significant difference was found in intrinsic goal orientation in favor of the STEM-RAP students, who also had different contextual images of their mentors as 'research partners'. The mentor interviews revealed several interaction themes, including content, procedural, and epistemic knowledge development, partnership, and emotional support. The findings emphasize the importance of research activities as part of active learning pedagogy for developing students' motivation to study science.

**Keywords:** mentor–student interaction; scientific dispositions; STEM career choice; STEM research apprenticeship; active learning; secondary school

## 1. Introduction

For over a decade, considerable effort and resources have been invested in understanding the best practices, environments, an [1–3]) and subsequently choose STEM career pathways [4]. Science educators have recognized the value of experiences beyond formal curricula for introducing students to science and engineering practices [5,6]. This is especially important in the 21st century, considering the requirement from science educators to integrate into the curriculum different types of scientific knowledge and higher-order thinking skills [7–11].

Active learning encourages students to do things as opposed to passively listen to lectures [12,13]. It involves students' engagement in their own learning process rather than passively receiving information from a teacher or textbook [14,15]. Active learning is based on the constructivist approach for learning while students extract information from different points of view, draw connections to their own experiences, apply diverse learning styles, and reflect on the learning processes [16–19]. Inquiry-based learning or science research experience is a type of active learning in which students ask questions, collect and analyze data, draw conclusions, and construct their own understanding of a topic [20,21]. Rather than memorizing facts and formulas, active and inquiry-based learning encourage students to think critically and creatively, which can foster meaningful learning of STEM concepts and increase the motivation to choose and be interested in STEM [22,23]. Active

learning and inquiry-based learning can also promote problem solving skills, which are essential for success in STEM; by engaging in hands-on activities and asking questions, students learn how to approach and solve real-world problems [24]. This type of learning can also promote collaboration and communication skills, as students often work in pairs or teams and share their ideas and findings with their peers [25]. Both active learning and inquiry-based learning or STEM research projects can foster critical thinking, curiosity and interest in learning, which is of particular importance for students who consider pursuing careers in STEM fields [24]. Finally, some educators claim that active learning pedagogy can improve the education system by increasing equity via narrowing performance and achievement gaps between mainstream and underrepresented groups [26–28].

STEM research apprenticeship programs (STEM-RAPs) for secondary school students are one such unique experience that may be valuable for students with an initial interest in STEM [29–31]. In STEM-RAPs, secondary school students actively participate in authentic science or engineering research, where they are exposed to contextual influences for an extended period [32]. Previous studies have shown that students who participate in a brief apprenticeship experience can develop a sense of belonging to science e.g., [33]. Furthermore, studies on research apprenticeships and other undergraduate research experiences have indicated correlations between self-efficacy, scientists' identity, and undergraduates' desire to pursue careers in science [34,35]. However, further research is needed to link desired outcomes to specific aspects of this active learning experience and to highlight the aspects considered by diverse students [36–38]. One of the challenges that STEM reforms were designed to meet is the provision of meaningful, active, and productive learning opportunities that promote diversity, equity, and inclusion in STEM education [39–42]. Acknowledging and understanding the effect that demographic features, such as gender, ethnic group, and cultural background, may have on students' scientific dispositions, motivational beliefs, and behaviors related to science, are critical for reducing gaps in enrollment and retention among different student populations [43–45]. This highlights the importance of examining the impact of the secondary school STEM-RAP on participants' scientific dispositions, STEM career choice, and mentor–student interaction.

Long-term active learning processes offer both students and educators an opportunity to better express and capture their scientific dispositions. Assessing the scientific dispositions of students who chose to cope with these unique learning and mentoring experiences can deepen the understanding of each aspect and the interface between them [46] and improve the effect of the program as a meaningful learning opportunity.

The goal of this study was to investigate the impact of a secondary school STEM research apprenticeship program (STEM-RAP) as part of active learning pedagogy on students' performance.

### 1.1. Background

#### 1.1.1. Apprenticeship as a Long-Term, Complex, Authentic Inquiry

The nature of scientific knowledge and processes, also known as scientific inquiry or practices, has evolved over time, as have the ways of teaching and learning science [9]. However, science educators continue to discuss the place of inquiry in the teaching and learning of science, its importance [47,48], and how inquiry can promote the choice of a STEM career [49–51]. This is also true for STEM-RAP, in which learners participate in an ongoing authentic project with an unknown answer to a problem or question that is relevant to the STEM community [52]. Apprenticeship programs typically take place in authentic, professional contexts on college campuses or in research institutes. This kind of setting fosters expert-apprentice relationships and gives students unique insights into and understanding of how scientists think and function [53,54].

However, investigating the effect of research in secondary schools is an area beyond the topics covered in standard national exams and demands considerable effort and emphasis on independent, active, and authentic learning [7,55]. The students engage with an up-to-date scientific issue, and the research process is intended to express their knowledge and



inquiry or research skills. Approval of the research proposal is required before it begins. As in the case of professional scientists, the end product of the long-term, complex inquiry is an academic, scientific write-up of the research and results. In its entirety, this could be the most authentic experience in which a secondary school learner can participate [56]. Coping with challenges during research can lead to an ongoing process, in which long-held beliefs and behaviors can be altered by more recent experiences [57].

### 1.1.2. Scientific Dispositions

Learning dispositions, or "learners' orientation toward learning", which drives them to behave in a certain way [58] p. 93, plays an important role in complex and authentic learning environments. Dispositions relate to the discipline being taught and the level at which learners identify with it and develop a sense of affiliation [59–61]. The model for measuring scientific disposition developed in this study is based on theoretical features of motivational beliefs and motivated behaviors [62]. Motivational beliefs include self-efficacy—individuals' subjective perceptions of their capabilities of doing a task well [63,64]. Motivated behaviors include individuals' choices and engagement [62,65]. Scientific dispositions are thus a multifaceted construct composed of several features [66,67], including self-efficacy in science, intrinsic goal orientation, and a sense of learning science. Fostering students' scientific dispositions requires the cultivation of their confidence and persistence in developing the skills they need to successfully perform tasks individually and cope with the challenges involved in conducting research.

### Self-Efficacy in Science

Bandura's social cognitive theory defined self-efficacy as people's judgments of their ability to produce designated levels of performance [63]. Perceived self-efficacy in an academic domain has been found to predict sustained effort, choice, and performance in that domain [68]. Academic choices and academic performance are expected to be correlated with the belief about one's ability to perform different academic tasks. Examples include effective writing and speaking, critical thinking, analysis of quantitative problems, using computing and information technology, working independently, or working in a team [69].

### Intrinsic Goal Orientation in Science

Researchers of motivation distinguish between intrinsic and external motivation [70]. Individuals may direct their actions toward pursuits of internal sources, such as personal interest, enjoyment, and learning [71]. Intrinsic motivation is a motivated behavior aimed at achieving a sense of ability and control over a situation [72]. Previous studies e.g., [73,74] indicated positive correlations between intrinsic motivation and the three psychological needs suggested by [75] for competence, relatedness, and autonomy. Learning experiences can facilitate or forestall intrinsic motivation by supporting or undermining the three psychological needs [76].

### Sense of Control over Learning

The motivation to engage in independent learning is a significant measure of learning engagement because independent learners take control over their learning for improving the results of their academic achievement. Such motivation affects learners' behavior in challenging situations and even in failure that they may experience during research [77].

### STEM Career Choice

The identities and career goals of adults are influenced by their early experiences and expectations at home and in school [78,79]. These factors impact their self-efficacy, which, in turn, can explain their motivation to perform certain actions in academic and career contexts [63,80]. Self-efficacy and social cognitive career theory posit that self-efficacy and outcome expectations predict career interests and behaviors related to career choice [80].

The role of self-efficacy in the formation of specific career attitudes is well established, but it is likely that other factors are involved as well [81]. Individuals tend to choose careers in which they will be able to succeed. Career plans, decisions, and aspirations have an important role in motivating behavior [80]. A personal interest that is strongly linked to intrinsic motivation is the most meaningful psychological variable in making career decisions [82,83].

### 1.1.3. Apprenticeship as an Environment for Mentor-Student Interaction

Apprenticeship research programs for secondary school students offer a unique authentic experience related to the work of scientists. Theories of mentoring conceive it as a dynamic developmental process, in which the relationship and interactions between a mentor and a mentee change over time [84]. To understand such relationships, researchers have investigated several aspects, such as the characteristics of effective mentors, the outcomes of the mentoring, and the roles of cultural and contextual variables in mentoring interactions [85,86]. While mentors listen, share their own story, provide feedback, give advice, and refer mentees to useful resources, they take on a number of roles [87,88]. Mentor relationship theory, based on [89], identified two broad classes of mentor functions: (1) career functions, which are mentor behaviors that foster mentee career development and advancement by modeling, coaching, and providing feedback, and (2) psychosocial functions, which foster mentee psychological and social development by interacting with mentees on a personal level to enhance their self-efficacy, sense of identity, and overall job comfort through emotional support [69].

Most empirical studies of apprenticeship research programs that have examined mentor–student interaction in secondary school or undergraduate research programs considered the positive outcomes of participating in apprenticeship research, including the student's future interest in STEM [29,31,57,90–92]. Research has also indicated an association between self-identity, science identity, and a student's desire to choose a career in STEM [34,35] on one hand, and the impact of mentors' intentions to promote active learning during the research experience [93] on the other hand.

In the present study, we examined student and mentor perceptions related to student–mentor interactions as demonstrated in secondary school apprenticeship research. The goal of the study was to investigate the impact of active learning pedagogy, represented by a secondary school STEM research apprenticeship program (STEM-RAP), on students' (a) scientific dispositions, which include self-efficacy, intrinsic goal orientation, and sense of control over science learning, (b) STEM career choice, and (c) interactions with their scientific mentors. In a comparison between students who participated in the STEM-RAP program and those who did not, the constructs of scientific dispositions, STEM career choice [2,21], and forms of interactions between students and mentors [86] were examined.

## 2. Materials and Method

In this study, we used convergent-design mixed methods.

### 2.1. Research Questions

The following research questions guided our study:

1.  What are the scientific dispositions and STEM career choices of secondary school students who participate in a STEM-RAP? Are there differences according to different subgroups (by gender and ethnic group)?
2.  What differences in scientific dispositions, if any, are there between students who participate in a STEM-RAP and those who participate in other STEM programs?
3.  Which aspects of mentor-student interaction are evident in the STEM-RAP experience as perceived by the students and mentors?

To address the three research questions, we applied two tools: (1) a questionnaire was used with its closed- and open-ended parts serving for the quantitative and qualitative aspects, respectively; and (2) interviews with a sample of mentors were conducted. We com-

pared the quantitative statistical results with the qualitative findings to better understand the research problem by triangulation and drawing conclusions [94].

*2.2. Description of the Program*

The STEM-RAP was integrated into the school curriculum and supported throughout the country by science centers and informal organizations, which provided access to research facilities, universities, and industrial research labs. It differed from other school or out-of-school research apprenticeship programs, such as research summer camps, in two main respects: at the beginning of the program, students were required to submit a research proposal for approval by the Ministry of Education according to scientific and pedagogical requirements. At the end of the process, submitting a final scientific research paper, similar to a mini thesis, was required of each student. This paper had to include a literature summary and synthesis, presentation of findings accompanied by analysis, and conclusions. These parts were reviewed by an independent academic mentor (advisor), who was an expert in the subject matter. Upon completion of the research work and a follow-up examination, each participating student was awarded extra credits on their matriculation certificate—a certificate of completion of the standard national matriculation examinations. The science centers at the various Israeli academic institutions led a training program that includes academic scientific writing, inquiry techniques in the laboratory, and assigning personal mentors for each student to guide his/her scientific research process. The guidance of each scientific mentor is in accordance with his/her educational domain of study, perceptions toward science and scientific experience and according to the Israeli Ministry of Education milestones set for submitting drafts of the research work.

*2.3. Participants*

The research population included 262 11th and 12th grade secondary school students studying science or technology as a major and taking part in the STEM-RAP. The program took place in several academic institutions in Israel. Participation in the program is based on the choice of the students who come from different schools to the academic institution closest to their area of residence. The students were engaged in several different research fields as part of STEM-RAP: biology ($n = 124$), physics ($n = 33$), chemistry ($n = 26$), multi-disciplinary studies ($n = 26$), engineering ($n = 21$), mathematics ($n = 10$), and other domains ($n = 22$). An example of a research topic in biology was Characterization and understanding of the key to activating the immune system—Calcinurin-NFAT protein track. Additionally, 57 students who were enrolled in a STEM secondary school major and conducted a smaller research project responded to the same questionnaire. The overall sample was representative of different types of schools, neighborhoods, and ethnic communities.

Table 1 presents the characteristics of the participants. The participants in this study are students who completed their projects. The overall attrition rate was about 30%, with most of the dropouts occurring at the preparation stage, i.e., after choosing a topic and submitting a proposal but before beginning the actual research experience.

**Table 1.** Characteristics of the research participants—11th–12th grades.

| Section | Description | STEM-RAP Students (n) | STEM-Major Students (n) |
|---|---|---|---|
| Gender | Male | 118 | 17 |
| | Female | 144 | 40 |
| Community [1] | Jewish | 215 | |
| | Arab | 47 | |

[1] Data regarding the community/culture (Jewish/Arab) was collected only among students who participated in the STEM-RAP for responding to the first research question.

Of the 12 mentors who were interviewed, eight were men and four women. Three of them were professors, five had PhDs, and the rest had a master's degree or a BSc degree in engineering.

### 2.4. Research Tools

An open- and closed-ended questionnaire was designed for this study to assess the participants' scientific dispositions, STEM career choice, and mentor-student relationship. The closed-ended questionnaire, summarized in Table 2, included two sections, which were completed at four stages of participation in the STEM-RAP: (a) at the beginning of the first year of the program during 11th grade, (b) at the end of the first year, i.e., the end of the 11th grade, (c) after the summer camp in the second year at the beginning of 12th grade, and (d) at the end of the program, after submitting the scientific research paper, at the end of 12th grade. A comparison group of 57 secondary school students, who were also STEM majors but did not participate in the STEM-RAP, responded to the same questionnaire.

**Table 2.** Questionnaire sections, categories and items examples.

| Section | Category (# of Statements) | Item Example | Cronbach Alpha Reliability |
|---|---|---|---|
| Section 1 | Self-efficacy—Science (5) | If I learn in a way that suits me, I will be able to cope with any new material in the sciences. | 0.624 |
| | Intrinsic goal orientation (3) | In scientific subjects, I prefer learning material that will stimulate my curiosity, even if it is hard to learn. | 0.584 |
| | Sense of control over learning (4) | I am confident that I will be able learn the skills that are required in the sciences. | 0.592 |
| | STEM career choice (5) | I would like to continue with higher education in the sciences and engineering. | 0.832 |
| Section 2 | Scientific relationship (4) | Work with a scientist/engineer increases my sense that science is an interesting field. | 0.715 |
| | Motivational- Affective relationship (4) | Working alongside a scientist/engineer raises my confidence to conduct scientific research. | 0.744 |

Cronbach's alpha values obtained in the study are defined as satisfactory according to Taber (2018).

The first part of the questionnaire asked for the participant's demographic and personal data, including gender, age, mother tongue, and scientific background. Section 1 then referred to the participant's scientific dispositions: self-efficacy in science, intrinsic goal orientation in science, and sense of control over learning. It was adapted from [46], Changes in Attitudes about the Relevance of Science (CARS) Questionnaire [37], and the Chemistry Career Choice (C3) questionnaire [1]. The students were also asked to rate their level of interest in engaging in a scientific-engineering field after their graduation on a scale of 1 to 5.

Section 2 of the questionnaire was designed to assess the students' perceptions on the mentoring relationship and examine their scientific and affective relationships with their mentors. The researchers formulated the items, which were then validated in an interview with two graduates of a STEM-RAP. The perception of the interaction between the mentor and the student was also assessed by a closed question, in which the students were asked to choose the image that best represented their student–mentor relationship out of three images: teacher–student, scientist–apprentice, and research partners. If none of these images were deemed fit, the student was asked to provide another image. The students completed Section 2 of the questionnaire when they were already in the program.

The questionnaire included three open-ended questions, each was a sentence that the student was asked to complete: (1) "Given the difficulty involved in the research, I think that...", (2) "While I was asked to work with my mentor, I felt that...", and (3) "Considering the skills I have acquired till now, I think that...".

One researcher compiled a rubric with the subjects' answers, then, each of the three researchers indicated for each relevant answer the scientific disposition reflected in it or

the tendency to choose STEM career. The researchers discussed their disposition choice for reaching a higher agreement. Interrater agreement was calculated by the number of concordant responses divided by the total number of responses multiplied by 100 (Altman, 1991). Analysis of the students' responses to these open-ended questions was conducted in two rounds by the three authors, reaching 85% interrater agreement (75% is the recommended minimum according to [95]).

### 2.5. Factor Analysis

The questionnaire responses were analyzed using exploratory factor analysis (EFA) for the closed-ended questions to identify adequate categories for analysis. EFA estimates factors that influence responses on observed variables [96,97]. Both EFA and principal components analysis (PCA) are used to reduce the number of parameters. In PCA, the variation in the component is based on the variation in item responses. In EFA, the variation of the items is based on the variation of a construct, so it was deemed most suitable for use in the present study, as we wanted to consider only shared variance. The oblique rotation method (Oblimin) was used, based on the assumption of correlations between the factors. The factor analysis was conducted for all statements, limiting the number of categories to five, as we designed the questionnaire based on relevant questionnaires in the literature (KMO = 0.827; Bartlett's Test = 0.000; $\chi^2(153) = 558.46$; Explained variance = 60%). All items were loaded onto one of the factors and the factor loading cutoff was 0.398. Examination of the initial categories detected by EFA yielded a good fit with the theoretical framework, so these categories, which are presented in Table 2, were adopted. The resulting alpha Cronbach (0.584–0.744) was similar to, but somewhat lower than that found in a previous study (0.792–0.862; [98]).

The goals of the interviews with the mentors were: identifying factors related to their choice how to guide the mentee's scientific work, characterization of the research process and scientific guidance, and characterization of aspects of the development of scientific thinking by the student. Examples of questions: Why did you choose to guide a high school student who conducted science research? Characterize your guidance process. Did you arrange the process in a certain structure? What instructional model have you used with your students? What type of questions were raised throughout the process by the learner, which reflected the development of his/her scientific thinking?

## 3. Results

In the first stage of the analysis, data were tested for normality. The Kolmogorov–Smirnov and Shapiro–Wilk tests were used for each dependent variable separately and in all cases the *p*-values were less than 0.05. Since the data were not normally distributed, non-parametric tests were used.

### 3.1. Scientific Dispositions and STEM Career Choices during Research Apprenticeships

Figure 1 describes the statistics of the students' scientific dispositions and STEM career choice at different stages of the program during high school and after they graduated.

The results indicate high scores among the STEM-RAP participants in self-efficacy in science, intrinsic goal orientation, and sense of control over learning. Similarly, their intentions to choose a STEM career were also high. Using Mann–Whitney U tests to compare means by gender and ethnic group revealed no differences between female and male students in any of scientific dispositions or STEM career choice, but the sense of control over learning was weaker among members of ethnic minorities, with borderline statistical significance and a small effect (U(132) = 1295, z = −1.96, *p* = 0.005, r = −0.17). The results of the Kruskal–Wallis H test demonstrated differences in scientific dispositions at different points in time in self-efficacy in science (H(2) = 13.33, *p* = 0.001) and in STEM career choice (H(2) = 14.12, *p* = 0.001). Mann–Whitney U tests for any two of the three stages showed that there were fewer positive attitudes in the middle stage of the students' apprenticeship research project and more positive attitudes at the final stages of the project

regarding self-efficacy in science (U(111) = 949, z = −3.45, *p* = 0.001, r = −0.33) and STEM career choice (U(111) = 1000, z = −3.12, *p* = 0.002, r = −0.30). In both cases, a moderate effect was found.

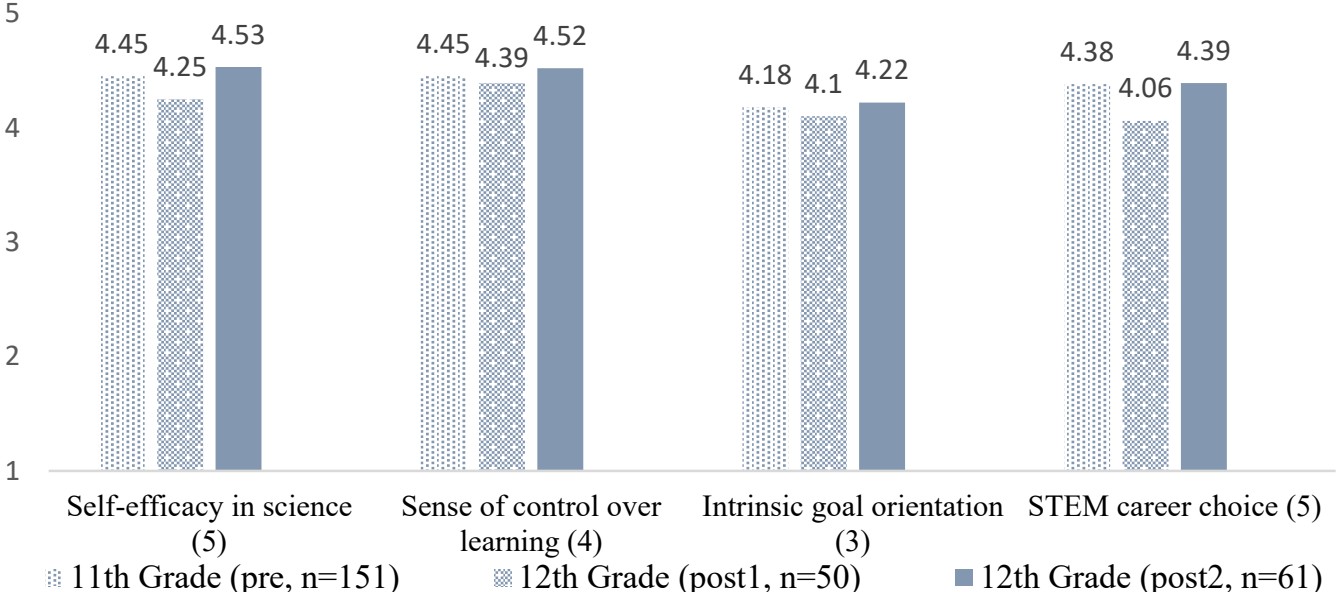

**Figure 1.** Students' STEM-RAP scientific dispositions and STEM career choices at different time points.

Analysis of the students' responses to the open-ended questions yielded several quotes that support the quantitative findings. For example, of the 115 expressions used by the twelfth-grade students, 40% were related to intrinsic goal orientation: "*I think that the main thing that is important to me in all my choices in learning science is the interest and the challenge*" [Student 11]. Of the 115 expressions, 36% related to the sense of control over learning: "*If I do not succeed in a particular subject, then it is entirely my responsibility, and I have to bear the consequences and do whatever is required*" [Student 13], and 24% related to self-efficacy "*Given the difficulty involved in the research, I think that I will succeed in my future academic studies*" [Student 7].

The relatively high level of participants' intentions to engage in science in the future was reflected also in the qualitative analysis of the open-ended questions in terms of the academic and professional aspects of their future. Most of the responses referred to the contribution of the program to their academic future in the field of STEM. Examples include: "*The research helps me understand what I want to learn in science in the future.*" [Student 21]; "*I feel ready for academia. It would not require me to develop a way of thinking that I did not have before the program.*" [Student 73]; "*Before the program I wanted to learn something different and not in the science field, but after entering this program I feel that I have the ability to study and also succeed in this field in the future.*" [Student 85]. Examples regarding the contribution of the program to the students' professional future in the field of STEM include: "*The program presents the world of science outside of theoretical studies. It will help me in getting a job in the field*" [Student 26], and "*It is significant to me because it is my future—what I have planned for my future—a science profession*" [Student 63].

To better understand the scientific dispositions of the STEM-RAP participants, we examined the differences between them and the students who were STEM high school majors but did not participate in the program. We included in this comparison only 12th grade students after considering the previous findings presented in relation to the changes measured between the groups of students throughout the program. Figure 2 shows a descriptive statistical comparison between STEM-RAP and non-STEM-RAP participants. Mann–Whitney U test results indicated no statistical differences between the two groups.

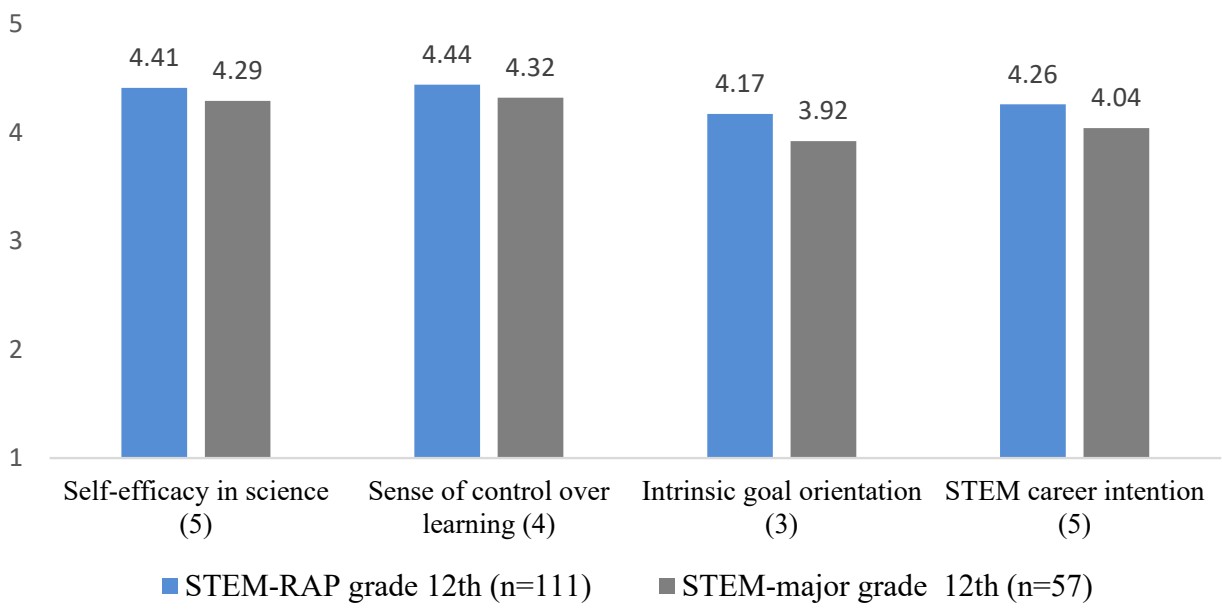

**Figure 2.** Scientific dispositions and STEM career choices of STEM-RAP and STEM majors.

### 3.2. Mentor–Student Interaction

To examine the mentor–student interactions as perceived by the students, we performed a quantitative analysis of their responses at various stages of the program. We focused on two types of relationships: the scientific relationship, which contributes to the development of knowledge and skills, and the motivational–affective relationship, which is characterized by feelings and emotions about working with the academic mentor. Figure 3 presents the results of the descriptive statistics. With respect to attitudes towards the scientific relationship, no statistical differences were found among the different stages of the program using a Kruskal–Wallis H test. In contrast, the Kruskal–Wallis H test revealed that there were significant differences in the perceived motivational-affective relationship (H(2) = 5.56, *p* = 0.018) at different points in time.

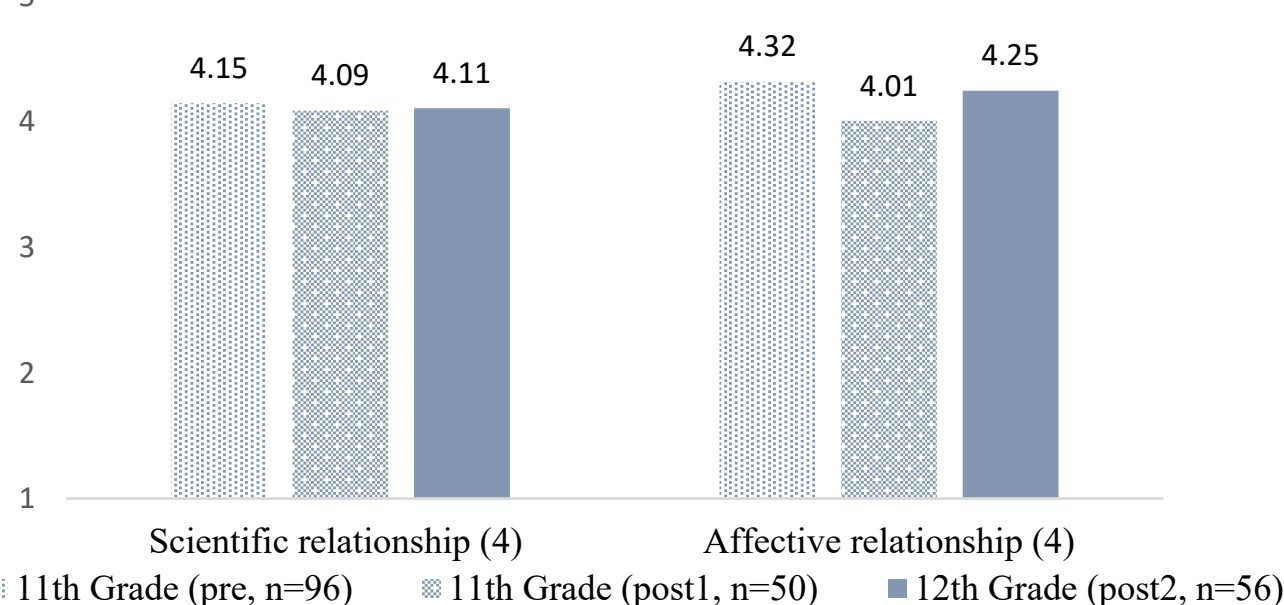

**Figure 3.** Perceived relationship with the scientific mentor at different stages of the program.

Mann–Whitney U tests for any two of the three stages showed that the motivational-affective relationship scored lower among students at the beginning of 12th grade (post-1) compared with the beginning of the program (pre) with a small effect (U(89) = 693, z = −2.36, *p* = 0.018, r = −0.25).

We also analyzed the image of the relationship between the scientific mentor and the student that the students selected to describe this relationship. Figure 4 presents the results of the descriptive statistics.

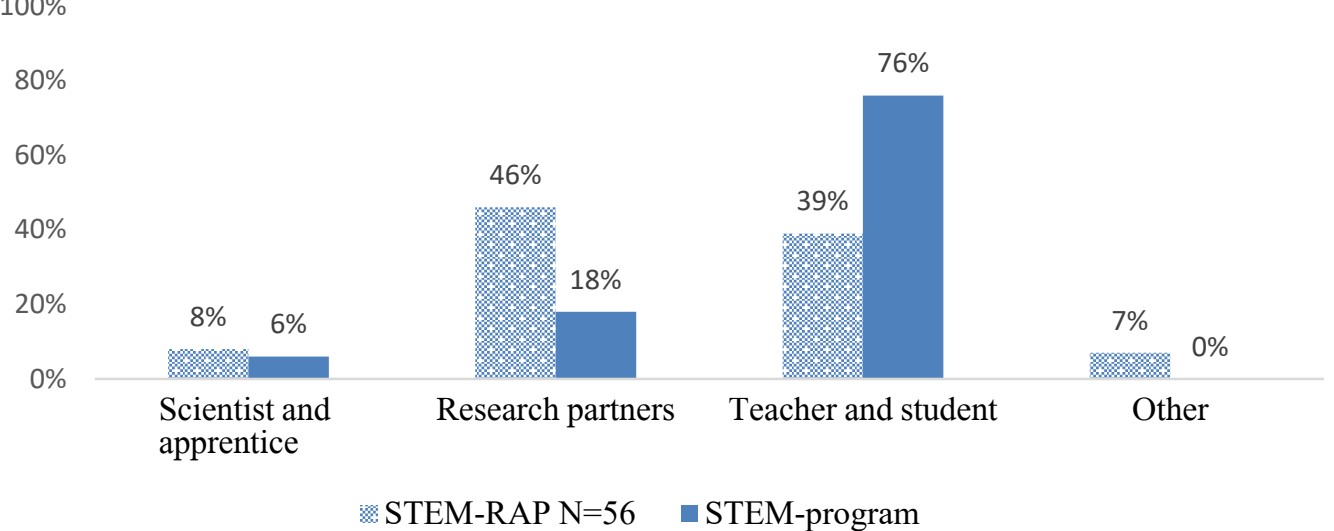

**Figure 4.** The image students selected to represent their student-mentor relationships.

Analyzing the responses of the STEM-RAP and STEM high school major students using a chi-square test indicated significant statistical differences between the two groups ($\chi^2$ = 13.762, *p* < 0.005, Cramer's V = 0.391). The significant differences were in two images. Among the STEM-RAP students, who experienced guided scientific research, 46% perceived their relationship with the mentor as "research partners" compared with 18% among STEM major students. In contrast, 76% of STEM major students perceived the relationship with the mentor as a teacher–student relationship, compared with 39% of the STEM-RAP students. In the category of "other", student answers included "*instructor and trainee*" and "*a friend who involves and shares*".

Comparison of the different types of mentor–student relationships among STEM-RAP students by gender, ethnic group, and area of residence using Mann–Whitney U tests yielded results similar to those regarding the scientific dispositions and STEM career choice. No significant differences were found between male and female students in their perceived cognitive and affective relationships with the mentors (see Table 3a). In contrast, comparison of the majority and minority ethnic groups revealed statistically significant differences in terms of the scientific relationship with the mentor (with small effect), with no differences in the affective relationship (see Table 3b).

Analysis of the students' responses to the open-ended questions revealed several categories with statements that supported the quantitative findings and conclusions. All the statements refer to the necessity and contribution of working with the mentor.

Regarding the scientific contribution, students said: "I feel that today I know how to write an academic paper on a scientific subject. The mentor helped me a lot in writing the paper and understanding the scientific subject." (Student 4); "Working with a scientist gave me a lot because my supervisor had tremendous knowledge and it was very good for me." (Student 55); "It helped me examine things more scientifically." (Student 52); "It showed me the world of science outside theoretical studies. It has to be recognized not only for final exanimation credit but also for work experience." (Student 82).

**Table 3.** Students' attitudes towards their relationships with their mentors, by gender. b. Students' attitudes towards their relationships with their mentors, by ethnic group.

| Category/(No. of Items) Min = 1, Max = 5 | Male $n = 92$ Mean (*S.D.*) | Female $n = 105$ Mean (*S.D.*) | Mann–Whitney U Test |
|---|---|---|---|
| Scientific–cognitive relationship (4) | 4.08 (0.70) | 4.18 (0.83) | n.s. |
| Motivational–affective relationship (4) | 4.20 (0.64) | 4.24 (0.64) | n.s. |
| | | (**a**) | |

| Category/(No. of Items) Min = 1, Max = 5 | Majority $n = 158$ Mean (*S.D.*) | Minority $n = 40$ Mean (*S.D.*) | Mann–Whitney U Test |
|---|---|---|---|
| Scientific–cognitive relationship (4) | 4.06 (0.80) | 4.40 (0.61) | U(89) = 505, $z = -2.23$, $p = 0.026$, $r = -0.24$ |
| Motivational–affective relationship (4) | 4.19 (0.67) | 4.33 (0.56) | n.s. |
| | | (**b**) | |

Students' statements on the affective contribution included: "I feel that it was interesting and challenging" (Student 16); "I feel that it contributed to my self-confidence" (Student 21); "There was sometimes a big gap between my supervisor and myself in scientific knowledge and in understanding each other." (Student 54); "The matching of mentor and student to conduct research is correct and necessary. I learned from our deep conversations." (Student 63); "The thesis is not only mine, but a collaborative product, I could only have reached this result with the help my supervisor." (Student 64).

Analysis of the interviews with the mentors revealed three central themes related to the mentor–student interaction: knowledge development, partnership, and emotional support. The knowledge development theme was divided into three categories: content knowledge, procedural knowledge, and epistemic knowledge. Table 4 summarizes the results.

**Table 4.** Themes and categories in mentor-student interactions from interviews with mentors.

| Quotes | Category | Theme |
|---|---|---|
| *"Every year I had two or three students that I guided in the research process and started with lectures. I gave them a series of lectures on semiconductors and devices." (mentor 2)"I start the research process with the theory, to see how they perceive and understand the concepts." (mentor 4)"They always received information (about the principles of the method), they went home, came back and then we would discuss this issue together." (mentor 2)* | Content knowledge | |
| *"Then the students prepared the bacteria, everything from scratch. It got to the point where one student was completely independent, I demonstrated to her, and she did the experiment completely by herself" (mentor 6)"I think the student I guided was special and I don't think this is a representative experience. I sent him a protocol or an explanation regarding what needs to be done and within fifteen minutes he already understood everything" (mentor 8)"In the first sessions, the students actually prepare and learn the background of the field, in terms of knowledge so that they understand what they are doing" (mentor 3)"In the second stage of the research process I move to the practical tools, in our case it was bioinformatics on the computer" (mentor 4)"The projects of two of my students in biology and biotechnology were more complex from a technical point of view, so they required more close guidance [in performing the experiment]" (mentor 6)* | Procedural knowledge | Knowledge development |
| *"When we got to the discussion section, we talked about how we should write the discussion..., we need to see if the claim we are making is consistent with what the scientific world is or if it is a contradictory claim. We need to show the contradictions, or we need to show papers that support, or papers that tested what we tested under similar conditions... They see that it is necessary to refer to the literature" (mentor 7)* | Epistemic knowledge | |

**Table 4.** *Cont.*

| Quotes | Category | Theme |
|---|---|---|
| I said to him [the student]: *"You heard what we do in the lab, go home and think about what you want to investigate that is related to what we are investigating and I will see if I can help you with it. He [the student] came back with a general idea and together we discussed the idea and formulated a research question"* (mentor 8)*"One student was very independent in her thinking. It's not like I told her that the experiment would have B, A and C, what do you think? She had a very high comprehension ability, she was brilliant. There were initiatives that were entirely her own that suited the research exactly as I thought. Again, not with all students it's 'straight forward'"* (mentor 7) | | Partnership |
| *"The technical parts of the process are less important—more successful, less successful... what is important is that you see a student who is trying, and you help him a lot because you want to promote him"* (mentor 3)*"Encouraging a student who goes home nervous is very important, because he should not go that way. The understanding that there is not always success is an important part of the process. We've all experienced it"* (mentor 9)*"One of my students did not come to the meetings, at first we only met once every three months. I called him to meet, and he said he had other occupations in sports—in football. Although the whole environment wanted him to progress (the school, his mother...) he was not interested enough. I thought he wouldn't finish the research, but I managed to pull him in and we did great research and he got a high score. It was a miracle"* (mentor 2)*"I had one student who broke down towards the end of the research process. I had to pick her up and encourage her and help her more with writing as well"* (mentor 1) | | Emotional support |

## 4. Discussion

The goal of this study was to investigate students' scientific dispositions and their intentions to choose a STEM career as they were expressed during and after their apprenticeship experience, and what role, if any, gender and ethnic origin play. The STEM-RAP was chosen as an example of the application of active learning pedagogy in the context of the high school system.

Secondary school STEM-RAPs were investigated previously, generally focusing on these programs' potential for and effectiveness in promoting participants' positive attitudes towards and interests in science and science learning, and choice of a science career [36,45]. The majority of the programs studied were short-term out-of-school frameworks, and the investigation concerned whether or not the programs were generators of key educational experiences that affected participants' adult decisions and actions [20,21,99]. Our study examined a different perspective of these experiences. We assessed the characteristics of learners that are relevant to successful participation in the scientific research process. Specifically, we examined the scientific dispositions of students who chose to participate in an 18-month research apprenticeship. By understanding the participants' motivational beliefs and motivated behaviors and how they were expressed during this unique learning experience, we sought to highlight the factors involved in STEM-related career pathways. A premise of our study was that the participants were probably interested in STEM studies before enrolling in the program Therefore, the core of this study was the measurement of the participants' scientific dispositions, their STEM career choice, and the scientific and motivational–affective aspects of mentor–student interaction during the different stages of the research apprenticeship experience.

Regarding the first research question, we found, as expected, that the scientific dispositions were high. Students who chose to participate in the program were characterized by a high degree of self-efficacy, intrinsic goal orientation, and sense of control over learning. These findings are consistent with previous results regarding research apprenticeship experiences [31,100]. However, differences in scientific dispositions between the different points in time were demonstrated only in self-efficacy in science and STEM career intentions; these were expressed in the middle stage and increased by the end of the program. Several studies have indicated that apprenticeship programs helped students confirm and clarify their decisions to pursue scientific careers [92,101]. In the current study, the students were exposed to a realistic perspective of scientists' work and products. They gained understanding of what it means to conduct scientific research, perform complex laboratory

experiments based on a high level of scientific thinking, cope with open-ended research questions, and complete advanced reading and writing tasks. This may explain the changes in the self-efficacy in science and STEM career choice found among some of the participants. The differences that were found regarding sense of self-efficacy and career choice intentions during the program are consistent with those of [92], who documented changes in scientific inclinations among groups engaged in different educational programs and emphasized the educational significance of changes in attitudes.

One of the challenges in STEM study is to provide meaningful and effective active learning opportunities for diverse students [14,39–41,102]. Our hypothesis was that participation in a research apprenticeship can increase self-efficacy among a variety of students and influence the motivated behavior driven by this experience. The literature has consistently reported fewer positive attitudes to STEM domains among female compared with male students. In our study, we did not find statistically significant differences between boys and girls in scientific dispositions or career choice. These results may support earlier findings of a reduction in the gender gap in sciences and engineering [44].

One limitation of this study is that not all the 11th graders who responded to the questionnaires also responded to the questionnaires in the 12th grade. Therefore, the effect of the program on students' self-efficacy in science, STEM career intentions, and scientific and motivational–affective relationship with their mentors in the middle stage and final stages of the study should be further explored in other countries and cultures.

Conducting research in partnership with an academic mentor can contribute to strengthening the active learning experience and therefore the positive attitudes toward STEM, as this study has shown. Similar to findings on the gender gap, some studies have documented fewer positive attitudes toward science among students from ethnic minorities than their counterparts from the majority ethnic group. However, other research has suggested that technologically rich programs, as well as those tailored to community and culture, contribute to positive attitudes towards science and the development of a science identity among students from of ethnic minorities [39,102]. The present research indicated no effects of ethnic group on the aspects of scientific dispositions examined. An explanation for the positive attitudes among students from the minority ethnic group may be related to their integration into unique programs designed for minority students. Several researchers have argued that special programs for minorities and women have an important impact on the advancement of these populations [21,39,40,102,103]. Furthermore, researchers and policy makers have argued that students must be prepared and inspired if they are to enter a career in the STEM fields, especially when it comes to members of underrepresented groups [81].

In respect to the second research question, in comparing the scientific dispositions between students who participated in the STEM-RAP program and those who did not, the Mann–Whitney U tests results indicate no statistical differences between the two groups. One explanation for not finding differences between the two groups may be related to the fact that the values of the scientific dispositions measured in the current study were found to be very high among all 12th grade students who chose to specialize in the field of STEM regardless of their participation in the STEM-RAP program. Another explanation may be related to the time of participation in the program. The findings may emphasize the time required to achieve changes in scientific dispositions. The achievement of educational outcomes among students following a pedagogical intervention has been reported to take between one and a half to three years [104,105].

Regarding the third research question, the findings reveal different understandings regarding the student–mentor interactions. Like the differences that were found between the different stages in the program in self-efficacy in science and STEM career intentions, the motivational–affective relationship scored lower among students at the beginning of 12th grade (post-1) compared with the beginning of the program (pre). This relationship scored higher among those who had already submitted their research papers (post-2) compared with the middle point of the program (post-1). In addition, more STEM-RAP

students, who experienced guided scientific research, perceived their relationship with the mentor as "research partners" compared with STEM major students. In contrast, more STEM major students perceived the relationship with the mentor as a teacher–student relationship, compared with the STEM-RAP students. These findings point to a more collaborative research type of relationship between the mentors and the students in the STEM-RAP program.

Comparison of the cognitive student–mentor relationships of male and female students revealed no significant differences, even though most of the mentors were men. These results can be linked to previous research findings that female students perform better when they are able to express their knowledge and thoughts in writing via open-ended questions in an interactive environment that enhances collaboration with others or encourages creativity and openness [20]. Accordingly, the close work with a mentor during the STEM-RAP can be considered as a factor that narrowed the gender gap. In the present study we also found that students who were members of an ethnic minority perceived the scientific relationship with the mentors more positively than students who were members of the majority ethnic group did. This supports the conclusions of [106] that mentoring junior researchers serves as a supportive anchor and promotes the science skills of those who are underrepresented in STEM professions. These findings are reinforced by the interviews with the mentors, whose analysis revealed four central themes of interactions: knowledge development—content, procedural, and epistemic, as well as partnership and emotional support.

*Contributions and Recommendations*

The research has a limitation related mainly to the number of participants (especially in the comparison group) and it is important to expand the research among larger groups of students. Despite this limitation, the study combines quantitative and qualitative tools and examines various aspects of the STEM-RAP program. Therefore, it offers insights into the promotion of choosing a career in STEM by means of student apprenticeship in authentic research and participation in active learning during secondary school. It highlights the importance of the research environment in science and engineering education, especially for the development of motivational factors in learning science and choosing STEM career pathways. Both active learning and inquiry-based learning may encourage students to study science and engineering subjects, be engaged, curious and motivated to learn. These students' characteristics can be particularly crucial and effective for students who are planning to pursue careers in STEM fields [2,107]. The research apprenticeship program provides an opportunity to change the role of the educator from an authoritative teacher to a mentor who leads his or her students in the active learning and knowledge-building processes [16] and to foster new interactions between mentors and students in STEM education.

We recommend further investigation of the effect of similar apprenticeship research programs on students' self-efficacy in science, STEM career intentions, and on their scientific and motivational–affective relationships with their mentors in order to reaffirm and strengthen the conclusions and extend the study to other countries in order to be able to generalize the findings and conclusions for other contexts.

Science educators should be aware of the decline in students' positive attitudes toward science and conducting scientific research in the middle stage of their two-year research project, which later recovers. Thus, they should intensify the support and encouragement of their students to carry on with their projects in order to decrease the attrition rate. Lastly, we did not find gender differences, and we attribute this finding to the design of our STEM-RAP program, which focused on active learning pedagogy, the apprenticeship research process, and on the one-on-one interaction between scientists or engineers and the students. We therefore advocate emphasizing these characteristics of the program to narrow the gender gap.



**Author Contributions:** Conceptualization, Y.J.D. and I.S.; methodology, Y.J.D. and I.S.; validation, Y.J.D. and I.S.; formal analysis, I.S. and M.E.; investigation, Y.J.D., I.S. and M.E.; resources, Y.J.D., I.S. and M.E.; data curation, M.E.; writing—original draft preparation, M.E.; writing—review and editing, Y.J.D. and I.S.; supervision, Y.J.D. and I.S.; project administration, Y.J.D.; funding acquisition, Y.J.D., All authors have read and agreed to the published version of the manuscript.

**Funding:** This research was funded by Chief Scientist, Ministry of Education grant number 23/7.17.

**Institutional Review Board Statement:** The study was conducted in accordance with the Declaration of Helsinki, and approved by the Behavioral Sciences Ethics Committee at the Technion in 2017 and the Chief Scientist of the Ministry of Education in Israel #9545.

**Informed Consent Statement:** Informed consent was obtained from all subjects involved in the study.

**Data Availability Statement:** Data available on request from the authors.

**Conflicts of Interest:** No author reported any financial or other conflicts of interest in relation to the work described.

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
