# Peer review of "Secondary School Apprenticeship Research Experience: Scientific Dispositions and Mentor-Student Interaction"

_education, doi:10.3390/educsci13050441_

Round 1

Reviewer 1 Report

This is an interesting study to investigate the impact of active learning in secondary school STEM research apperentship program, at scientific disportins, STEM career choices and inetractions with mentors. 

A more precise explanation of the main constructs is necessary for this otherwise well-presented research. The survey lacks the context of the participants. Are all the students from the same school, what are the size of the school(s) and in which country was the survey carried out?   

I would like to know more of what did the 12 mentors been interviewed about.

Author Response

Thank you for the important comments. Attached a file with our reference to the comments and the description of the changes made in the paper (marked red in the revised file).

Reviewer 2 Report

This manuscript is well constructed and very well written. 

Author Response

Thanks for the positive feedback

Reviewer 3 Report

Please see the file attached.

Author Response

(The authors gave the same response as above.)

Round 2

Reviewer 3 Report

I appreciate the authors' efforts and find the manuscript to be much improved now. I have only two further comments:

1. In my first review I suggested to move the research questions into a respective section. Now they have been moved to the end of the introduction which I find highly unfitting. The research background should motivate and thus lead to the research questions so that after posing the research goal the methods can be introduced.

2. The reporting of the Mann-Whitney U-test statistics should be checked again. If they are reported according to APA, the number in parantheses indicates the number of observations. In some cases, this number is 2 (line 352) or even 1 (346) which does not make sense.

Author Response

Thank you for the comments. Following, our reference to the two comments -

  1. At the end of the introduction, we deleted the research questions and replaced them with the research goal, to help the reader in understanding the reasoning for the sections in the theoretical background. At the beginning of the materials and methods we added and modified a few sentences and included a sub-section, titled: Research questions.
  2. We corrected the reporting of the Mann-Whitney U-test 
